# Early Diagnosis in Prader–Willi Syndrome Reduces Obesity and Associated Co-Morbidities

**DOI:** 10.3390/genes10110898

**Published:** 2019-11-06

**Authors:** Virginia E. Kimonis, Roy Tamura, June-Anne Gold, Nidhi Patel, Abhilasha Surampalli, Javeria Manazir, Jennifer L. Miller, Elizabeth Roof, Elisabeth Dykens, Merlin G. Butler, Daniel J. Driscoll

**Affiliations:** 1Division of Genetics and Genomic Medicine, Department of Pediatrics, University of California, Irvine, CA 92868, USA; juneannegold@gmail.com (J.-A.G.); nidhipatel.1992@gmail.com (N.P.); surampallia@gmail.com (A.S.); jmanazir@uci.edu (J.M.); 2Children’s Hospital of Orange County, Orange, CA 92868, USA; 3Health Informatics Institute, University of South Florida, Tampa, FL 33612, USA; Roy.Tamura@epi.usf.edu; 4Department of Pediatrics, Loma Linda University Medical School, Loma Linda, CA 92350, USA; 5Department of Pediatrics, University of Florida, Gainesville, FL 32610, USA; millejl@peds.ufl.edu (J.L.M.); driscdj@peds.ufl.edu (D.J.D.); 6Vanderbilt Kennedy Center for Research on Human Development, Vanderbilt University, Nashville, TN 37203, USA; elizabeth.roof@Vanderbilt.edu (E.R.); elisabeth.m.dykens@Vanderbilt.edu (E.D.); 7Departments of Psychiatry & Behavioral Sciences and Pediatrics, University of Kansas Medical Center, Kansas City, KS 66160, USA; mbutler4@kumc.edu

**Keywords:** Prader–Willi syndrome, age diagnosis, obesity, deletion, uniparental disomy

## Abstract

Prader–Willi syndrome (PWS) is an imprinting genetic disorder characterized by lack of expression of genes on the paternal chromosome 15q11–q13 region. Growth hormone (GH) replacement positively influences stature and body composition in PWS. Our hypothesis was that early diagnosis delays onset of obesity in PWS. We studied 352 subjects with PWS, recruited from the NIH Rare Disease Clinical Research Network, to determine if age at diagnosis, ethnicity, gender, and PWS molecular class influenced the age they first become heavy, as determined by their primary care providers, and the age they first developed an increased appetite and began seeking food. The median ages that children with PWS became heavy were 10 years, 6 years and 4 years for age at diagnosis < 1 year, between 1 and 3 years, and greater than 3 years of age, respectively. The age of diagnosis and ethnicity were significant factors influencing when PWS children first became heavy (*p* < 0.01), however gender and the PWS molecular class had no influence. Early diagnosis delayed the onset of becoming heavy in individuals with PWS, permitting early GH and other treatment, thus reducing the risk of obesity-associated co-morbidities. Non-white individuals had an earlier onset of becoming heavy.

## 1. Introduction

### 1.1. Clinical Aspects of Prader–Willi Syndrome

Prader–Willi syndrome (PWS) affects about 1/15,000 individuals and is characterized by the lack of expression of genes on the paternal chromosome 15, located in the 15q11.2–q13 region [1,2,3]. The majority of imprinted genes in this region are involved in both RNA and protein processing of neuroregulators and hormones at the brain level. Disruptions in these genes negatively affect neuronal development, endocrine function and hormone levels, leading to the PWS phenotype [4,5,6,7]. Clinical features in the neonatal period include poor tone and suck, hypogonadism, feeding difficulty and failure to thrive. Later findings include a characteristic facial appearance, early-childhood onset of excessive hunger (hyperphagia), which can lead to morbid obesity if uncontrolled, mild intellectual disability, growth and other hormone deficiencies, leading to a short stature and small hands and feet, along with a distinctive behavioral phenotype, with temper tantrums, outbursts and self-injury (skin picking) [5,6,8]. Obesity related complications include cardiovascular problems, diabetes mellitus, hypertension, sleep apnea, gastric distension, necrosis, and choking as causes of death [4,9,10,11,12]. The diagnosis of PWS is often delayed, leading to excessive medical costs, parental anxiety and increased time before treatment with, e.g., growth hormone (GH) [13]. GH therapy in PWS allows for increased stature, muscle mass, strength and physical activity, thereby improving metabolic rate and energy expenditure, resulting in decreased fat mass and obesity status, particularly when administered at a young age [4,14,15,16,17,18,19]. When given at a young enough age, it improves the muscles used in sucking and feeding, enabling the avoidance of gastric tube placement. Although cognitive benefits of GH treatment have been identified in animal models and other patients with GH deficiencies, such ancillary effects of GH treatment in PWS have not been well studied. However, Dykens et al [20] showed the cognitive and adaptive advantages of early and continued GH treatment, and children with PWS who began treatment before 12 months of age had higher Nonverbal and Composite IQ scores than children who began treatment between 1 and 5 years of age. Most recently, Butler et al. [21] reported significantly higher IQ scores in the Vocabulary section of the Stanford–Binet test in the GH treated group when compared with non-GH treatment. These studies further emphasize the importance of earlier diagnosis and initiating treatment quickly. 

### 1.2. Genetic Aspects of Prader–Willi Syndrome

Prader–Willi syndrome is a complex disorder of genomic imprinting caused by three main mechanisms, which ultimately results in a complete absence of paternally expressed genes in the 15q11.2–q13 region. The three PWS molecular genetic classes include a paternal deletion of the 15q11.2–q13 region (61% of cases), maternal uniparental disomy (UPD) 15 (36%), and an imprinting defect (ID) at 3% [22,23]. In the PWS chromosome region, the paternal gene copies are typically expressed, while the maternal alleles are silenced, due to a parent-of-origin specific imprinting process involving DNA methylation and other epigenetic factors during gametogenesis. The diagnosis of PWS is traditionally based on clinical suspicion and confirmed by a DNA methylation testing of chromosome 15, which detects 99% of individuals with PWS [4,6,24]. 

Driscoll et al. (2017) describes a comprehensive testing strategy to establish the specific genetic mechanism of an individual with DNA methylation analysis consistent with PWS [24]. Hartin et al. [25] provides an updated approach, using a genetic testing flow chart for PWS. The DNA methylation specific PCR (mPCR) test is the most rapid and cost-effective method to date in diagnosing PWS, however, it does not determine the specific PWS molecular classes. Chromosomal microarray analysis with SNP probes is currently the best method for identification of the individual PWS molecular classes, since it detects individuals with paternal 15q11–q13 deletions, segmental and total maternal isodisomy, and microdeletions of the imprinting center, however additional genetic testing is required in about 15% of patients in whom microarray results do not identify the genetic defect; the latter often requires parental DNA samples and chromosome 15 genotyping [22]. Methylation Specific -Multiplex Ligation-dependent Probe Amplification (MS-MLPA) of chromosome 15 and non-chromosome 15 polymorphic DNA markers may also be used to detect PWS once the diagnosis is under consideration [25,26,27]. In a study done by Bar et al. [28], the most common causes of delayed diagnosis in PWS were due to clinical features being missed in the neonatal period, as well as the use of fluorescence in situ hybridization (FISH) analysis for testing, which was the preferred method before the availability of methylation-specific PCR. FISH will only identify the 15q11–q13 deletion, which accounts for 60% of individuals with PWS, not maternal disomy 15 or imprinting defects. Next generation sequencing using chromosome 15 probes and, possibly, methylation-specific quantitative melting point analysis (MS-QMA) of imprinted genes in the chromosome 15 region, may become viable techniques in the future, to allow for more accurate and cost-efficient measures for early diagnosis, possibly including newborn screening [29,30].

## 2. Materials and Methods

The NIH-sponsored Rare Diseases Clinical Research Network (RDCRN) of the PWS/ Early Onset Morbidity (EMO) dataset was developed during the period 2008–2014, and data from the network were used in our study. Initially, the dataset was developed for natural history studies, characterization of diagnostic and therapeutic plans, and genotype–phenotype correlations in PWS. The RDCRN dataset has been utilized for several publications to date, which focused on the molecular and natural history, as well as the clinical characterization of PWS [5,25,31,32,33,34]. 

Our analysis was conducted on data collected from individuals with Prader–Willi syndrome recruited for the RDCRN Natural History 5202 protocol and stored at the Data Management Coordinating Center at the University of South Florida (Tampa), as described by Butler et al. [35]. The dataset included 355 individuals with genetically confirmed Prader–Willi syndrome, 37% of whom were diagnosed after the age of one year and 25% after the age of three years (with ages ranging from 1 month to 48 years). The age of diagnosis was elicited from the Natural History Form and approved by the local IRB Committees from the four participating clinic sites located in California, Kansas, Tennessee and Florida. If the data entry point was missing, then the age of diagnosis was defined as the age at which the last genetic test was performed on the enrolled subject.

### Analysis

Three variables from the Natural History Form were analyzed: Age when child was first reported to become heavy; Age at which increased appetite first developed and Age the child first started to seek food. The age the child first became “heavy” (e.g., at or above the 85th percentile for weight for age and gender) was determined with input from the primary care providers and historical records for the majority of subjects enrolled. The majority of subjects between ages 2–20 years, who first became heavy during the trial, had a BMI above the 85th percentile for age and gender. 

The variables were analyzed by the Cox proportional hazards model and by developing Kaplan–Meier curves, as undertaken in other reports on PWS (e.g., [36]). The earliest reported age for each of these variables was used as the endpoint age for each variable. Subjects who did not indicate an age for the variable were considered censored at their last recorded age. Because of the large variability in age at diagnosis (ranging from approximately 1 month to 48 years), the age of diagnosis was categorized into three categories, based on age distribution (<1 year, ≥1 years and <3 years, and ≥3 years) with the majority (62%) diagnosed at <1 year and 26% diagnosed at ≥3 years. In addition to age of diagnosis (categorized), ethnic background (white vs. non-white), gender, and PWS molecular class were included as covariates in the Cox hazard model and analyzed for statistical significance.

## 3. Results

Summaries of the number of subjects with PWS in each of the three age categories (<1 year, >1 years and <3 years, and ≥3 years) and other covariates are shown in Table 1. Kaplan–Meier curves for the age when the child first became heavy as determined by their primary care providers, age the child developed an increased appetite, and age the child began to actively seek food are shown in Figure 1, Figure 2 and Figure 3, respectively. The Cox proportional hazards analyses for these three variables are shown in Table 2. For example, individuals with PWS having an imprinting defect had a hazard ratio (HR) of 1.45 of becoming heavy, compared to UPD15 or 15q11–q13 deletion molecular genetic classes, but the number of subjects tested with imprinting defects was low. The 15q11–q13 deletion group had the highest HR of 1.31 for an increased appetite, as well as for actively seeking food, with an HR of 1.14.

Both the age of diagnosis (*p* < 0.001) and race (*p* = 0.004) were significant factors influencing the age when the child was first reported to be heavy. The earlier the diagnosis of PWS, the later the age at which individuals became heavy. The estimated median age for when the child first became heavy was 10 years for an age of diagnosis of < 1 year, 6 years for an age of diagnosis between 1 and 3 years, and 4 years for an age of diagnosis greater than 3 years. Additional partitions of age of diagnosis ≥ 3 years category were also examined, but no evidence was found that further partitioning into separate categories produced significant statistical differences. Non-white individuals became heavier at an earlier age, compared with whites, with an estimated median age of 4 years for non-whites and a median age of 8 years for whites. However, age of diagnosis and race did not influence the age at which individuals first developed an increased appetite or began actively seeking food. In addition, our data analysis indicates that the age the individual became heavy, age of increased appetite and age of seeking food were not significantly different across the three PWS molecular classes (deletion, UPD or imprinting defects), except for a difference between the 15q11–q13 deletion and UPD15 subjects regarding increased appetite (Figure 1, Figure 2 and Figure 3).

## 4. Discussion

Early diagnosis of Prader–Willi syndrome, particularly in the newborn period, is critical for changing the lives of those with this disorder and further supported by our study results, showing that those with earlier diagnosis developed obesity at a later age. We firmly believe that early diagnosis in the first few weeks of the newborn period is critical for individuals with PWS to receive appropriate intervention and anticipatory guidance. This should happen at an early an age as possible. For example, GH treatment at an early age affords the opportunity to take proactive strategies for regulating caloric intake and delaying the onset and reducing the risk of early obesity, with associated co-morbidities such as diabetes, hypertension, and respiratory compromise. The earlier diagnosis would also impact the cost of medical care, by decreasing diagnostic evaluations and the length of hospital stays, as noted by Shoffstall et al. [13]. 

The typical time of diagnosis for individuals with PWS was noted approximately three decades ago to be between 7 and 9 years of age, depending on the genetic subtype (deletion vs. non-deletion status [8]). In our cohort of 352 analyzed subjects with PWS, the age of diagnosis in our younger subjects had decreased to a mean age of 3.1 years, but often not enough to avoid extensive and costly evaluations, with health concerns leading to ineffective medical care and treatment. We believe that it is critical to detect those with PWS in the newborn period, in order for treatment to begin as early as possible. Significant progress has been made in awareness and early diagnosis of PWS, but further efforts could be made to diagnose at earlier stages. 

Better acceptance of expanded newborn screening programs nationwide regarding metabolic and genetic disorders may impact this problem. Early diagnosis and treatment can significantly improve prognosis in other disorders not readily detected at birth by routine physical examination, but sensitive, specific, inexpensive tests do exist, using expanded newborn screening programs with filter paper cards, a gold standard for newborn testing and diagnosis of genetic disorders [2]. As an example, on 21 May 2010, the Secretary of Health and Human Services added Severe Combined Immunodeficiency (SCID), an immune disorder with a frequency of 1/53,000 to the Recommended Uniform Newborn Screening Panel (RUSP) [1,3]. Pompe disease, another rare metabolic disorder was also added to the newborn screening list in several states. Since the clinical presentation and available treatment of PWS meets or fulfills the criteria for newborn screening, it is expected that newborn screening for PWS will become available in the future, depending on cost-effective genetic testing methods, where early diagnosis can impact medical care, treatment and quality of life [37]. Early diagnosis can also transform medical management of PWS, by eliminating extensive and expensive evaluations, along with the uncertainty generated by not having a diagnosis early in infancy. If the diagnosis is not made early, the patient is deprived of the benefits of optimal treatment and anticipatory strategies to avoid morbid obesity. 

The benefits of lifelong GH therapy in infants, children and adults with PWS have been demonstrated in multiple well-designed and controlled studies [14,38,39,40,41,42,43]. For example, GH treatment for 2 years in children showed major increases in height and weight, and a decrease in body fat [44]. GH replacement therapy also improves linear growth velocity and, ultimately, height, and results in healthier body composition (increased lean body mass, decreased fat mass), muscle function and level of activity [4,40,45]. GH treatment in children with PWS ultimately improves growth, adult height and body composition, and nearly normalizes stature by 18 years of age, with a significant improvement in obesity status, as noted in PWS-specific standardized growth charts [10,18,46]. Evidence further supports treating PWS adults with GH, with it leading to increased muscle strength and physical activity, improved lipid levels and better quality of life measures after one year of treatment (e.g., [40]). The improvements with GH treatment are also demonstrated in bone mineral density [47,48]. When treatment occurs from infancy, facial appearance and body habitus also normalizes in conjunction with good dietary management, and there is an improvement in quality of life and psychosocial status in PWS individuals [45,49]. The benefits of initiating treatment before the age of 2 years are well recognized, and further improvement are possible, when the diagnosis and treatment is earlier particularly in the newborn period [50]. Early diagnosis, good dietary control, exercise and GH treatment with better therapeutic approaches [46,51] can reduce the risk and age of onset of obesity, and many of the associated co-morbidities, such as diabetes, hypertension, and respiratory compromise, common in PWS without early recognition and treatment. Despite significant advances in diagnosing PWS, the mean age of diagnosis is still delayed. As seen in our current study, there was a wide range of age at diagnosis, spanning birth–48.0 years. (mean ± SD = 3.1 yr. ± 6.7 yr.; median = 0.3 yr.). 

We believe it is important for individuals to be diagnosed in the newborn period, to receive better treatment and appropriate, syndrome-specific medical care, beginning as early as the first few weeks of life. The current genetic testing methods (e.g., mPCR, high resolution chromosomal microarrays) which are useful for diagnosing PWS in the newborn period, would also have the added benefit of detecting the majority of newborns with Angelman syndrome (AS), as well. AS is caused by a maternal chromosome 15q11–q13 deletion, whereas PWS is associated with a paternal 15q11–q13 deletion. AS is associated with severe intellectual disability, electroencephalographic (EEG) abnormalities and epilepsy, limited or absent language development, an abnormal gait, inappropriate laughter and autistic behaviors, with a frequency of 1/12,000 [52,53]. The combined frequency for the two genomic imprinting disorders of PWS and AS would be about 1/6000. This frequency is more common than nearly all other disorders for which newborn screening is currently available. The benefits for early detection for Angelman syndrome would also be substantial, as early diagnosis avoids the unnecessary diagnostic odyssey (and expense), as seen in PWS, and anxiety that families experience prior to an accurate diagnosis, permitting early therapy with anticonvulsants and interventions with support. Early diagnosis, identifying abnormal DNA methylation, which has a 99% accuracy rate for the diagnosis of PWS and a 78% chance of accuracy identifying AS, would also allow for the detection of imprinting defects for both PWS or AS, which can be associated with a 50% recurrence risk, thereby permitting early and accurate genetic counseling [22,24,52]. Large-scale newborn screening programs for PWS/ Angelman syndrome would also give us a much more accurate frequency of the disorder, which may be more prevalent than we previously thought. 

In summary, early diagnosis could lead to significant improvements, with decreased costs and better medical care of affected newborns (in both PWS and AS), leading to an enhanced quality of life. More research is needed to further investigate the feasibility of lowering costs of testing, including DNA methylation analysis and its application in the newborn setting.

## Figures and Tables

**Figure 1 genes-10-00898-f001:**
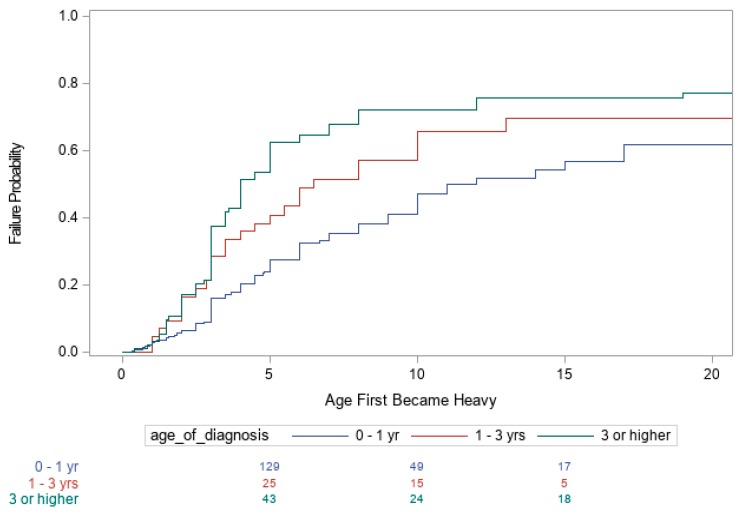
Kaplan–Meier Plot of the age individuals first become heavy.

**Figure 2 genes-10-00898-f002:**
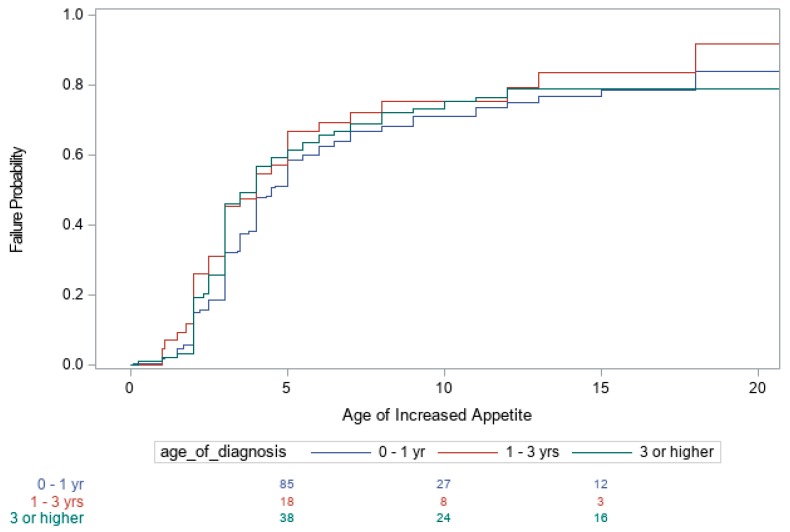
Kaplan–Meier Plot of ages individuals first developed an increased appetite.

**Figure 3 genes-10-00898-f003:**
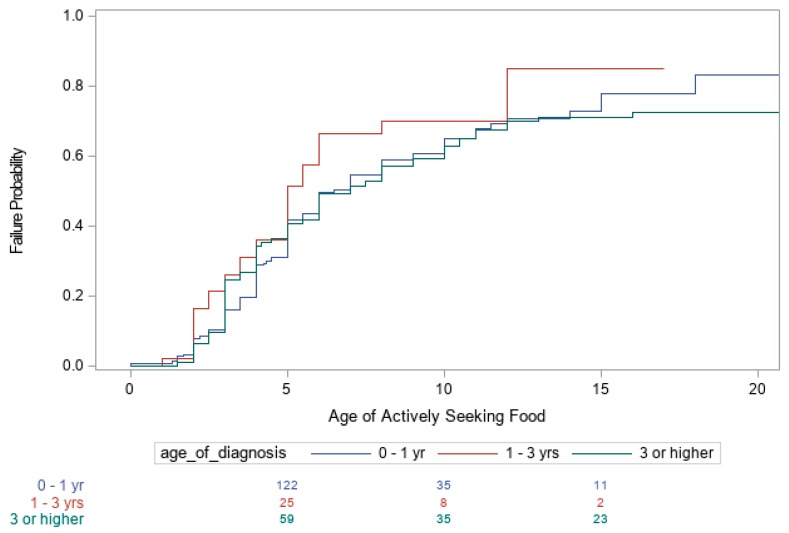
Kaplan–Meier Plot of the age individuals began to actively seek food.

**Table 1 genes-10-00898-t001:** Summary and frequency of Prader–Willi subjects in various categories.

Category	Frequency (%)
Age of Diagnosis (yrs.)	
Mean = 3.1	
Median = 0.3	
SD = 6.7	
Min =0.0	
Max = 48.0	
Age of Diagnosis Category	
<1 yr.	217 (62%)
≥1 yr. and <3 yrs.	42 (12%)
≥3 yrs.	93 (26%)
Gender	
Female	194 (55%)
Male	158 (45%)
Ethicity	
White	328 (93%)
Non-White	24 (7%)
Prader–Willi Molecular Class	
Deletion	216 (61%)
Imprinting Defect	11 (3%)
Uniparental Disomy	125 (36%)

**Table 2 genes-10-00898-t002:** Cox proportional hazard analyses for age at which individuals first becoming heavy, age of increased appetite, and age individuals began actively seeking food.

	First Becoming Heavy	Increased Appetite	Actively Seeking Food
**Effect**	Hazard Ratio (95% CI) *p* value	Hazard Ratio (95% CI) *p* value	Hazard Ratio (95% CI) *p* value
**Gender (ref = female)**	0.99 (0.73, 1.33) 0.990	1.13 (0.88, 1.46) 0.332	1.09 (0.83, 1.44) 0.525
**PWS Molecular Class**			
**Deletion vs. UPD**	0.90 (0.66, 1.23) 0.499	1.31 (1.00, 1.72) 0.054	1.14 (0.85, 1.52) 0.393
**ID vs. UPD**	1.45 (0.69, 3.07) 0.326	0.72 (0.33, 1.58) 0.415	1.05 (0.50, 2.20) 0.893
**Ethnicity (ref = white)**	0.46 (0.28, 0.78) 0.004	0.76 (0.48, 1.21) 0.179	0.73 (0.44, 1.21) 0.224
**Age of Diagnosis**			
**<1 vs. 1–3**	0.67 (0.43, 1.04) 0.077	0.70 (0.47, 1.03) 0.067	0.72 (0.48, 1.09) 0.125
**<1 vs. >3 yrs.**	0.48 (0.35, 0.66) < 0.001	0.90 (0.67, 1.20) 0.456	1.05 (0.77, 1.43) 0.754

Bold represent labels for the material in the rows.

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
