# Peer review of "Early Diagnosis in Prader–Willi Syndrome Reduces Obesity and Associated Co-Morbidities"

_genes, 2019, doi:10.3390/genes10110898_

Round 1

Reviewer 1 Report

The manuscript contains interesting findings, obtained in a large cohort of patients with Prader-Willi, and suggests that a earlier diagnosis would help patients to obtain a better clinical improvement.

It would be interesting to also have a measure of the cognitive status and/or quality of life of patients according to the age of diagnosis.

As a minor point, I suggest to do not use the word "RACE" and substitute it with an other (i.e. ethnicity, ancestry).

Author Response

Response:  In this  natural history study the quality of life of patients was not measured. The cognitive status is being analyzed for another publication and is beyond the scope of this manuscript.

We have substituted 'ethnicity' or 'ethnic background' in the manuscript for 'Race'.

Reviewer 2 Report

This is a clinical/statistical study of very limited scope that nonetheless provides important evidence for the benefits of early diagnosis of PWS.  As the study is based on a clinical database, the only methods described in the paper are the statistical methods used to establish the effects of early diagnosis.  These are utilized appropriately, and overall the paper is well written and of professional quality.  I have no major opposition to publication of the study, but would like to see section 1.2 of the introduction clarified in minor ways prior to publication:

Line 73.  Reflex testing and the nature thereof may not be familiar to a non-clinical audience.

Lines 72-79.  This paragraph lists several types of tests but does not clearly articulate them as pertains to PWS diagnosis.  Presumably the main point is that different types of genetic or epigenetic causes require different molecular tests for confirmation, so perhaps this idea could be used as a starting point and then elaborated by the molecular class of the cause.

Line 82.  FISH studies are mentioned only as a cause for incorrect diagnosis, with no prior mention of their use in diagnosis.  A little more explanation would be helpful, e.g., presumably FISH analysis was a preferred method before the availability of methylation-specific PCR.

Author Response

Response: The text has been modified to add clarity to the testing strategy for Prader Willi syndrome.